# Model-Based Assessment of Elastic Material Parameters in Rheumatic Heart Disease Patients and Healthy Subjects

**Mary A. Familusi** [1,2,*] , **Sebastian Skatulla** [1,*] , **Jagir R. Hussan** [3], **Olukayode O. Aremu** [4,5,6] ,
**Daniel Mutithu** [4,5] , **Evelyn N. Lumngwena** [7] , **Freedom N. Gumedze** [8] **and Ntobeko A. B. Ntusi** [4,5,6,9]

1   Computational Continuum Mechanics Research Group, Department of Civil Engineering,
    University of Cape Town, Cape Town 7700, South Africa
2   South African DST-NRF Centre of Excellence in Epidemiological Modelling and Analysis,
    Stellenbosch University, Stellenbosch 7600, South Africa
3   Auckland Bioengineering Institute, University of Auckland, Auckland 1142, New Zealand;
    r.jagir@auckland.ac.nz
4   Division of Cardiology, Department of Medicine, University of Cape Town, Cape Town 7700, South Africa;
    armolu002@myuct.ac.za (O.O.A.); mttdan008@myuct.ac.za (D.M.); ntobeko.ntusi@uct.ac.za (N.A.B.N.)
5   Cape Heart Institute, Faculty of Health Sciences, University of Cape Town, Cape Town 7700, South Africa
6   Cape Universities Body Imaging Centre, Faculty of Health Sciences, University of Cape Town,
    Cape Town 7700, South Africa
7   School of Clinical Medicine, Faculty of Health Sciences, University of Witwatersrand,
    Johannesburg 2000, South Africa; en.lumngwena@uct.ac.za
8   Department of Statistical Sciences, University of Cape Town, Cape Town 7700, South Africa;
    freedom.gumedze@uct.ac.za
9   Intersection of Noncommunicable Diseases and Infectious Diseases, Extramural Research Units,
    South African Medical Research Council, Cape Town 7505, South Africa
*   Correspondence: fmlmar001@myuct.ac.za (M.A.F.); sebastian.skatulla@uct.ac.za (S.S.);
    Tel.: +27-844-800-872 (M.A.F.); +27-021-650-2595 (S.S.)

**Abstract:** Non-invasive measurements are important for the development of new treatments for heart failure, which is one of the leading causes of death worldwide. This study aimed to develop realistic subject-specific computational models of human biventricles using clinical data. Three-dimensional finite element models of the human ventricles were created using cardiovascular magnetic resonance images of rheumatic heart disease (RHD) patients and healthy subjects. The material parameter optimization uses inverse modeling based on the finite element method combined with the Levenberg–Marquardt method (LVM) by targeting subject-specific hemodynamics. The study of elastic myocardial parameters between healthy subjects and RHD patients showed an elevated stiffness in diseased hearts. In particular, the anisotropic material behavior of the healthy and diseased cardiac tissue significantly differed from one another. Furthermore, as the LVEF decreased, the stiffness and its orientation-dependent parameters increased. The simulation-derived LV myocardial circumferential and longitudinal stresses were negatively associated with the LVEF. The sensitivity analysis result demonstrated that the observed significant difference between the elastic material parameters of diseased and healthy myocardium was not exclusively attributable to an increased LVEDP in the diseased heart. These results could be applied to future computational studies for developing heart failure treatment.

**Keywords:** subject-specific modeling; rheumatic heart diseases; finite element modeling; elastic material parameters

## 1. Introduction

Rheumatic heart disease (RHD), which refers to the long-term cardiac damage caused by either a single severe or multiple recurrent episodes of acute rheumatic fever (ARF), results in progressive fibrosis of heart valves [1,2]. The appropriate directionality of blood

flow through the heart chambers is hampered by functional abnormalities brought on by changes in the matrix architecture and cellular components [3]. Heart failure or death may result from untreated valvular heart disease [1].

Computational methods for examining healthy and diseased heart tissue are increasingly used to better understand the heart and the pathophysiology of cardiovascular disease (CVD) [4,5]. Realistic computer-based cardiac simulations provide clinically immeasurable and preclinical information [6]. For example, the myocardial stress distribution affects cardiac remodeling, but such a distribution is not clinically assessed [5]. Cardiac biomechanical models provide detailed 3D deformation, stress, and strain maps that complement clinical data [5,7]. The finite element method (FEM), in combination with advanced simulation tools and new cardiovascular imaging modalities, can be used to analyze the ventricular wall stress–strain distribution in providing greater insight into the physiology of normal subjects and CVD patients and to predict responses to medical and surgical interventions [5,7–9]. In such models, it is essential to accurately use the in vivo elastic biomechanical parameters of the human myocardium to mimic cardiac mechanics properly. Otherwise, the stress–strain prediction would be overestimated or underestimated, which will lead to inaccurate diagnostic information [10]. Meanwhile, despite the wealth of clinical data available and the continuous enhancements in the complexity and accuracy of subject-specific models, significant work is still required to enable the translation of model-based elastic material parameter assessment to the clinic [11,12].

The reliability of a finite element model is dependent on its underlying constitutive model, which continues to be a significant challenge for the biomechanics community. Various constitutive models that assume the myocardium as a transversely isotropic material have been formulated, e.g., [13–17]. While myocardial biaxial tension tests serve as the foundation for transversely isotropic models, myocardial tissue has markedly varied resistance to simple shear stress in various planes, according to the findings of shear studies. This implies that cardiac tissue is an orthotropic material with unique material properties in the orthonormal planes of symmetry during diastole [18,19]. As a result, orthotropic models of passive myocardium that account for the unique material response in three mutually orthogonal planes have been developed [20–23].

In this study, the constitutive model that describes the myocardium as a nonlinear, orthotropic, and nearly incompressible hyperelastic material proposed by Usyk et al. [23] was considered. We made use of an inverse modeling approach that is based on the finite element method combined with the Levenberg–Marquardt method (LVM) to obtain subject-specific material parameters for elastic mechanical modeling of the heart. We examined the difference in the elastic material parameters of RHD patients and healthy subjects, correlated the elastic material parameters with clinical data, and investigated the effect of end-diastolic pressure (EDP) on the estimated material parameters. Using information from high-resolution cardiac magnetic resonance (CMR) and diffusion tensor imaging based on CMR (DTI), it is possible to generate a high-fidelity geometry of both ventricular chambers in RHD and control subjects. Our modeling method simulates cardiac function by calibrating the elastic material parameters of the left and right ventricles to correspond to the in vivo clinical ventricular volume data. An excellent agreement with the empirical Klotz EDPVR was obtained for all subjects.

## 2. Methods

### 2.1. Study Population

In this study, the CMR-tagged cine images and ECG-triggered segmented k-space gradient echo sequence with spatial modulation of magnetization (SPAMM) in orthogonal planes were used to obtain the images of 30 human hearts at the Cape Universities Body Imaging Centre (CUBIC), Faculty of Health Sciences, University of Cape Town (UCT), South Africa, and Groote Schuur Hospital. The UCT Faculty of Health Sciences Human Research Ethics Committee approval (REF: 686/2018) and patients' consent were obtained to conduct research on unidentified human data. Thirty human subjects were included in

this study: fifteen RHD patients and fifteen healthy controls. Participants excluded from this study were pregnant women, those below the age of eighteen years, critically ill people, and those who had severe cardiac failure and contraindications to CMR.

### 2.2. CMR Image Analysis

CMR image analysis for the ventricular ejection fraction (EF) was performed using the CVI42 (Circle Cardiovascular Imaging, Calgary, Alberta) software (https://www.circlecvi.com/, accessed on 14 February 2023). The short axis (SA) stack was used to plan manual contouring of the epicardial and endocardial borders of the LV in end-diastole and end-systole, providing the corresponding LV cavity volumes LVEDV and LVESV, respectively, which were used to calculate the stroke volume as SV = LVEDV − LVESV and the ejection fraction as LVEF = SV/LVEDV. A similar procedure was used in obtaining the right ventricular ejection fraction (RVEF).

The myocardial pre- and post-contrast readings with hematocrit (HCT) correction, typically taken 15 min after the administration of contrast, were subtracted to generate the extracellular volume (ECV) measurement. The formula for the ECV is shown below:

$$\text{ECV} \quad = \quad (1 - \text{HCT}) * \frac{\Delta R1_{myocardium}}{\Delta R1_{blood}} \tag{1}$$

where $\Delta R1_{blood} = 1/\text{native T1 post-contrast} - 1/\text{native T1 pre-contrast}$. T1 mapping is a CMR imaging technique used to characterize changes in myocardial tissue composition in terms of intracellular and extracellular compartments.

### 2.3. Geometric Segmentation and Finite Element Model Creation of The Biventricle

The 3D patient-specific anatomical biventricular models were created using Synopsys's Simpleware ScanIP (Synopsys, Mountain View, USA) (https://www.synopsys.com/simpleware.html, accessed on 14 February 2023) from tagged CMR images at the onset of diastolic filling corresponding to the end-systolic volume (ESV), as this is the configuration closest to a zero pressure state that can be measured in vivo [4,24]. The segmented heart models were meshed using linear tetrahedral finite elements using the GiD pre- and post-processing software (CIMNE International Center for Numerical Methods in Engineering) (https://www.gidsimulation.com/, accessed on 14 February 2023). The pipeline from CMR image segmentation to the meshed anatomical finite element model is illustrated in Figure 1. Model calibration was performed using an in-house C++ code called SESKA (http://www.ccm.uct.ac.za/, accessed on 14 February 2023), a finite-element-method-based structural analysis software that includes a module for whole-heart simulations (see [25] for details).

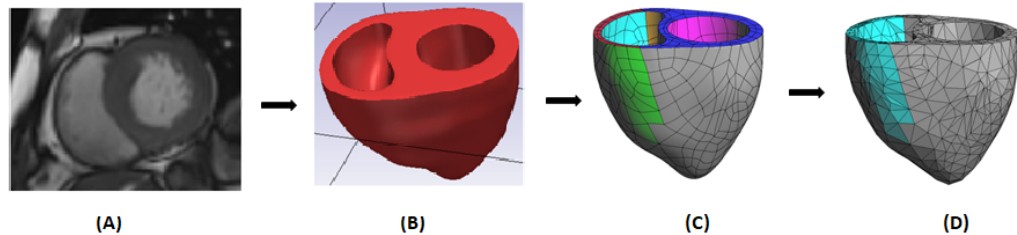

**Figure 1.** Construction of biventricular geometry from CMR images and the creation of layers in GiD to distinguish between LV and RV features (**A**) CMR short axis image. (**B**) Geometry obtained using Simpleware ScanIP. (**C**) Created layers in GiD: LV epicardium (grey), RV epicardium (green), LV endocardium (pink), RV endocardium (cyan), LV base (blue), and RV base (red). (**D**) The mesh associated with the segmentation.

### 2.4. Constitutive Model for Elastic Myocardium

In this study, the mechanical model proposed by Usyk et al. [23] and reformulated in terms of the invariants of the Green strain tensor, **E**, by Legner et al. [26] was considered for our human biventricular models. The model describes the myocardium as a nonlinear, orthotropic, and nearly incompressible hyperelastic material using the strain energy function given by:

$$\psi = \frac{A}{B}\left(\exp^{BQ_m} - 1\right) + A_{\text{comp}}\left(J\ln J - J + 1\right), \tag{2}$$

where parameters $A$ and $B$ are stiffness factors and $J = \det \mathbf{F}$ is the Jacobian, with $\mathbf{F}$ being the deformation gradient tensor, quantifying the volumetric deformation of cardiac tissue as linked to the numerical penalty parameter $A_{\text{comp}}$ enforcing the near incompressibility of the myocardium. Additionally, $Q_m$ is defined in terms of the Green strain tensor and the material directions defining structural tensors $\mathbf{M}_f$, $\mathbf{M}_s$, and $\mathbf{M}_n$ as follows:

$$
\begin{aligned}
Q_m := \ & a_1(tr(\mathbf{M}_f\mathbf{E}))^2 + a_2(tr(\mathbf{M}_s\mathbf{E}))^2 + a_3(tr(\mathbf{M}_n\mathbf{E}))^2 \\
& + a_4 tr(\mathbf{M}_f\mathbf{E}^2) + a_5 tr(\mathbf{M}_s\mathbf{E}^2) + a_6 tr(\mathbf{M}_n\mathbf{E}^2),
\end{aligned} \tag{3}
$$

where $a_i(i = 1, \ldots, 6)$ are the anisotropy coefficients associated with the three preferred material directions, namely fiber axis, $\mathbf{V}_f$, sheet axis, $\mathbf{V}_s$, and sheet normal axis, $\mathbf{V}_n$. These vectors construct an orthonormal basis and allow for the formulation of the so-called *structural tensor*:

$$\mathbf{M}_f = \mathbf{V}_f \otimes \mathbf{V}_f, \ \ M_s = \mathbf{V}_s \otimes \mathbf{V}_s, \ \ \mathbf{M}_n = \mathbf{V}_n \otimes \mathbf{V}_n. \tag{4}$$

The original parameters used in Usyk et al. [23] can be recovered by the following relations:

$$b_{ff} = a_1 + a_4 \tag{5a}$$

$$b_{ss} = a_2 + a_5 \tag{5b}$$

$$b_{nn} = a_3 + a_6 \tag{5c}$$

$$b_{fs} = \frac{1}{2}\left(a_4 + a_5\right) \tag{5d}$$

$$b_{fn} = \frac{1}{2}\left(a_4 + a_6\right) \tag{5e}$$

$$b_{sn} = \frac{1}{2}\left(a_5 + a_6\right). \tag{5f}$$

corresponding to

$$
\begin{aligned}
Q_m = \ & b_{ff}\, E_{ff}^2 + b_{ss}\, E_{ss}^2 + b_{nn}\, E_{nn}^2 + b_{fs}\left(E_{fs}^2 + E_{sf}^2\right) + b_{fn}\left(E_{fn}^2 + E_{nf}^2\right) \\
& + b_{sn}\left(E_{sn}^2 + E_{sn}^2\right).
\end{aligned} \tag{6}
$$

$b_{ff}$, $b_{ss}$, and $b_{nn}$ denote the axial stiffness parameters in the preferred material directions $\mathbf{V}_f$, $\mathbf{V}_s$ and $\mathbf{V}_n$, respectively, whereas $b_{fs}$, $b_{fn}$, and $b_{sn}$ are the corresponding shear stiffness parameters.

### 2.5. Boundary Conditions

To simulate the mechanics of the diastolic filling phase of the heart, it is necessary to apply appropriate boundary conditions. Two types of boundary conditions needed to be applied, the Dirichlet and Neumann boundary conditions. In terms of the Dirichlet boundary conditions, the ventricles were fixed at the base as shown in Figure 2A to prevent the heart from undergoing rigid body motion in the vertical direction and to allow for the connections between the base and the major blood vessels. The heart experiences a degree

of twist during muscular contraction; to allow for this torsional behavior, but still restrict the deformation to a physiologically normal level, another Dirichlet boundary condition was weakly imposed through the application of an elastic line force with spring constant $k = 0.1$ kN/mm acting in the tangential direction around the epicardial base, as seen in Figure 2B [26,27]. The elastic line forces effectively prevent rigid body motion in the short axis direction, but do not obstruct ventricular wall thickening. The elastic line force boundary condition is incorporated as the Neumann boundary condition:

$$\oint_{\partial \mathcal{C}_N} f_e \, \mathbf{c} \cdot \delta \mathbf{u} \, dS \tag{7}$$

with the curve $\mathcal{C} \subset \partial \mathcal{B}$ which is added to the weak form of the problem formulation as presented in [28]. $k$ denotes the elastic spring constant, $\mathbf{c} = \mathbf{F} \, \mathbf{c}_0$ the tangent vector on the epicardial base in the deformed configuration with $\mathbf{c}_0$ being its undeformed counterpart, $\delta \mathbf{u}$ the virtual displacement, $dS$ the differential line element of the undeformed epicardial base $\partial \mathcal{B}$, and $f_e = \sum_{i=1}^{n} k \, \Delta \mathbf{u}_i \cdot \mathbf{c}(t_i) / |\mathbf{c}(t_i)|$ the current magnitude of the elastic force, which has accumulated until simulation time step $t_n$ with $\Delta \mathbf{u}_i$ being the displacement increments of time steps $t_i$, $i = 1, n$.

The inflow of blood into the ventricular cavities during the diastolic filling stage of the cardiac cycle imparts hydrostatic pressure on the endocardial walls. To model this, an incremental surface pressure load was prescribed on the endocardium of both ventricles as the Neumann boundary conditions as shown in Figure 2C. The magnitude of the pressure increments was controlled by prescribed ventricular cavity volume increments [25]. Due to the lack of subject-specific ventricular pressure, which requires invasive measurements, we assumed the LVEDP as 3.0 kPa (22.50 mmHg) for diseased hearts and 2.0 kPa (15.00 mmHg) for healthy hearts [8,29,30]. One-third of LV blood pressure was applied to the RV endocardium [7,31].

The myocardial fiber orientation angles were assigned on the endocardium and epicardium, respectively, and subsequently, the three local preferred material directions $\mathbf{V}_f$, $\mathbf{V}_s$, and $\mathbf{V}_n$, respectively, were constructed using an algorithm developed by Wong and Kuhl [32] as illustrated in Figure 2D. The distribution of the preferred material directions throughout the myocardium of LV and RV was achieved by interpolation [33].

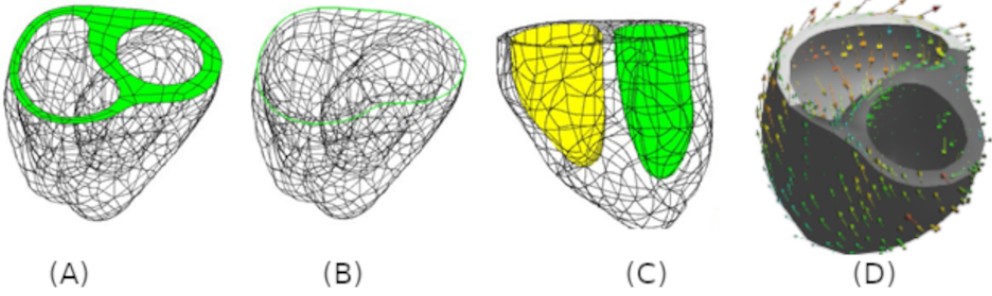

**Figure 2.** Dirichlet and Neumann boundary conditions: (**A**) The vertical displacement at the base of the heart was set to zero (fixed), to prevent heart movement. (**B**) An elastic boundary was placed at the base–epicardium interface with a stiffness value of 0.1 kN/mm. (**C**) Pressure values of 3.0 kPa and 1.0 kPa were assigned for the LV (green) and RV (yellow), respectively. (**D**) Myocardial fiber orientation angles are assigned on the endocardium and epicardium, respectively.

### 2.6. Parameter Optimization Procedure

The patient-specific estimation of the eight material parameters of the orthotropic constitutive law by Usyk et al. [23] and the epicardial and endocardial fiber orientation angles for the two ventricles of all thirty biventricular (BiV) models was performed combining the finite element (FE) method with the Levenberg–Marquardt algorithm (LVM) [27,34,35]. As optimization target served the analytical end-diastolic pressure volume relation (EDPVR)

curve by Klotz et al. [36], so-called *Klotz curve*. The Klotz curve was computed with the assumed EDP values for the LV and RV (see Section 2.5) Skatulla et al. [28] and utilized for the direct inverse computation of the unknown unloaded configuration of the heart ventricles ($V_0$) used as the reference volume so that the residual stress and strain associated with the corresponding end systolic pressure (ESP) was incorporated in the mechanical response of the BiV models.

　In line with the methodology proposed by Krishnamurthy et al. [37], the parameter optimization with respect to myocardial stiffness, anisotropy parameters, and fiber orientations was achieved by reducing the $R^2$ error between the Klotz curve and the simulated EDPVR curve of each subject, beginning with the least-loaded LV volume (ESV) and progressing until the end-diastolic volume (EDV) was reached at the stipulated end-diastolic pressure (EDP). The following initial values of the myocardial material properties were chosen: $A = 0.1$ kPa, $B = 1.0$, $a_1 = -6.0$, $a_2 = -5.0$, $a_3 = 9.0$, $a_4 = 12.0$, $a_5 = 12.0$, $a_6 = -6.0$, $A_{comp} = 100$ kPa, $\theta_{epi} = -57°$, and $\theta_{endo} = 72°$. The initial anisotropy parameters $a_i$, $i = 1 \ldots 6$ used in the strain-invariant representation of the orthotropic material law (Equation (2)) were taken from [33] and can be converted to the corresponding strictly positive parameters as originally proposed by Usyk et al. [23] using Equations (5a–f). The value of the numerical penalty parameter $A_{comp} = 100$ kPa was found most-suitable to to obtain near incompressible material behavior and avoid non-physiological deformation behavior and numerical instability for the chosen material model and finite element discretization [38,39]. Additionally, a sensitivity analysis was performed with regard to $A_{comp}$ in Section 3.7 to further substantiate the suitability of this choice. A preliminary parameter discovery search was undertaken within a parameter space based on [23] for $A = [0.015; 1.5]$, $\theta_{epi} = [-60°; -35°]$, and $\theta_{endo} = [60°; 83°]$ to obtain the above-stated initial values. The anisotropy parameters $a_i$, $i = 1 \ldots 6$ were not included in the preliminary discovery search due to inconclusive results. Instead, they were directly taken as given by [23] as initial values for the subsequent parameter optimization. The stiffness parameter $A$ was optimized for both ventricles individually to reflect patient-specific material behavior whereas the anisotropy coefficients $a_1$ to $a_6$ were only estimated for the LV and kept fixed for the RV due to the absence of clinical global strain data. Therefore, the scaling parameter $B$ had to be additionally determined for the RV to capture the highly nonlinear EDPVR, whereas for the LV, $B = 1.0$.

## 3. Results

### 3.1. Geometric Segmentation

　Using our inverse modeling approach, the unloaded left and right ventricular volume were obtained (see Tables 1 and 2). On average, the segmented and clinical end-systolic volume (ESV) for the RHD groups was 98.04 mL vs. 98.51 mL for the LV and 85.02 mL vs. 85.28 mL for the RV; they were 64.33 mL vs. 64.11 mL for the LV and 76.47 mL vs. 75.83 mL for the RV of the control group. The clinical and segmented ESV had a good fit, with a root-mean-squared error (RMSE) of 1.48 mL for the left ventricular end-systolic volume (LVESV) and 2.00 mL for the right ventricular end-systolic volume (RVESV) of the RHD patients. The RMSE values of 2.54 mL and 2.27 mL were obtained for the control LVESV and RVESV, respectively.

### 3.2. Statistical Analysis

　The median and interquartile ranges (IQRs) are reported for continuous variables, while categorical variables are presented as numbers (percentages) and were compared using the Chi-squared test [40]. The normality of the data was tested using the Shapiro–Wilk test [41]. For continuous variables, which were not normally distributed, a non-parametric test, the Mann–Whitney U test [42], was conducted to assess the difference between the RHD and control groups. A test of association between the continuous variables was performed using the Pearson correlation [43]. A significant linear correlation between two

variables is defined as both meeting a *p*-value $< 0.05$. All analyses were performed using Stata software Version 15 (College Station, TX, USA) [44].

**Table 1.** Comparison of the CMR and model cavity volumes at the end-systole, the end-diastole, and the unloaded volume ($V_0$) at pressure = 0 mmHg in RHD patients.

| | Left Ventricle | | | | | | | Right Ventricle | | | | | | |
|---|---|---|---|---|---|---|---|---|---|---|---|---|---|---|
| | $V_0$ | | ESV | | | EDV | | $V_0$ | | ESV | | | EDV | |
| | FEM | CMR | FEM | Error | CMR | FEM | Error | FEM | CMR | FEM | Error | CMR | FEM | Error |
| 1 | 35.0 | 37.0 | 35.1 | 5.2 | 90.0 | 90.0 | 0.0026 | 50.7 | 78.0 | 77.0 | 1.3 | 121.0 | 121.0 | 0.0002 |
| 2 | 53.0 | 51.0 | 53.5 | 4.9 | 135.0 | 135.0 | 0.0031 | 77.9 | 82.0 | 82.5 | 0.6 | 165.0 | 165.0 | 0.0018 |
| 3 | 108.9 | 111.0 | 110.2 | 0.7 | 235.0 | 235.0 | 0.0001 | 35.5 | 75.0 | 71.2 | 5.1 | 101.0 | 101.0 | 0.0003 |
| 4 | 75.5 | 77.0 | 78.5 | 1.9 | 178.0 | 178.0 | 0.0002 | 61.3 | 61.0 | 61.2 | 0.3 | 136.0 | 136.0 | 0.0020 |
| 5 | 110.6 | 109.0 | 111.5 | 2.3 | 256.0 | 256.0 | 0.0045 | 97.6 | 104.0 | 103.5 | 0.5 | 185.0 | 185.0 | 0.0140 |
| 6 | 135.9 | 144.0 | 143.7 | 0.2 | 243.0 | 243.0 | 0.0014 | 50.4 | 72.0 | 70.5 | 2.0 | 109.0 | 109.0 | 0.0007 |
| 7 | 83.1 | 90.0 | 88.7 | 1.4 | 168.0 | 168.0 | 0.0002 | 45.2 | 61.0 | 63.4 | 3.9 | 96.0 | 96.0 | 0.0021 |
| 8 | 94.9 | 101.0 | 101.9 | 0.9 | 189.0 | 189.0 | 0.0001 | 36.9 | 69.0 | 68.2 | 1.1 | 92.0 | 92.0 | 0.0003 |
| 9 | 82.3 | 86.0 | 86.0 | 0.0 | 170.0 | 170.0 | 0.0015 | 56.9 | 67.0 | 65.8 | 1.9 | 118.0 | 118.0 | 0.0005 |
| 10 | 137.8 | 141.0 | 143.9 | 2.0 | 292.0 | 292.0 | 0.0025 | 104.4 | 130.0 | 129.9 | 0.1 | 226.0 | 226.0 | 0.0006 |
| 11 | 118.8 | 119.0 | 120.1 | 0.9 | 301.0 | 301.0 | 0.0043 | 73.9 | 81.0 | 81.1 | 0.1 | 155.0 | 155.0 | 0.0011 |
| 12 | 129.8 | 146.0 | 145.6 | 0.3 | 210.0 | 210.0 | 0.0007 | 66.9 | 179.0 | 179.2 | 0.1 | 227.0 | 227.0 | 0.0010 |
| 13 | 76.0 | 97.0 | 97.2 | 0.2 | 136.0 | 136.0 | 0.0031 | 57.5 | 63.0 | 67.4 | 7.0 | 109.0 | 109.0 | 0.0012 |
| 14 | 68.4 | 80.0 | 79.2 | 1.0 | 132.0 | 132.0 | 0.0108 | 73.1 | 75.0 | 77.9 | 3.8 | 152.0 | 152.1 | 0.0448 |
| 15 | 77.5 | 82.0 | 83.1 | 1.3 | 157.0 | 157.0 | 0.0001 | 74.7 | 78.0 | 80.5 | 3.2 | 160.0 | 160.0 | 0.0010 |

CMR: cardiovascular magnetic resonance; FEM: finite element model; ESV: end-systolic volume; EDV: end-diastolic volume.

**Table 2.** Comparison of the CMR and model cavity volumes at the end-systole, the end-diastole, and the unloaded volume ($V_0$) at pressure = 0 mmHg in control subjects.

| | Left Ventricle | | | | | | | Right Ventricle | | | | | | |
|---|---|---|---|---|---|---|---|---|---|---|---|---|---|---|
| | $V_0$ | | ESV | | | EDV | | $V_0$ | | ESV | | | EDV | |
| | FEM | CMR | FEM | Error | CMR | FEM | Error | FEM | CMR | FEM | Error | CMR | FEM | Error |
| 1 | 79.0 | 78.0 | 79.8 | 2.3 | 203.0 | 203.0 | 0.0007 | 76.7 | 78.0 | 78.0 | 0.0 | 186.0 | 186.0 | 0.0001 |
| 2 | 53.4 | 54.0 | 54.3 | 0.5 | 125.0 | 125.0 | 0.0008 | 66.2 | 74.0 | 71.0 | 4.1 | 137.0 | 137.0 | 0.0008 |
| 3 | 56.2 | 54.0 | 57.6 | 6.7 | 123.0 | 123.0 | 0.0052 | 64.0 | 72.0 | 71.7 | 0.4 | 134.0 | 134.0 | 0.0007 |
| 4 | 57.5 | 57.0 | 58.0 | 1.7 | 154.0 | 154.0 | 0.0002 | 66.9 | 67.0 | 68.0 | 1.6 | 169.0 | 169.0 | 0.0001 |
| 5 | 68.5 | 71.0 | 69.7 | 1.9 | 162.0 | 162.0 | 0.0001 | 84.7 | 106.0 | 105.7 | 0.3 | 189.0 | 189.0 | 0.0006 |
| 6 | 49.5 | 50.0 | 50.1 | 0.3 | 122.0 | 122.0 | 0.0035 | 57.4 | 61.0 | 60.1 | 1.4 | 130.0 | 130.0 | 0.0017 |
| 7 | 60.4 | 63.0 | 63.3 | 0.5 | 129.0 | 129.0 | 0.0015 | 57.7 | 66.0 | 61.7 | 6.5 | 124.0 | 124.0 | 0.0011 |
| 8 | 46.4 | 49.0 | 47.0 | 4.0 | 117.0 | 117.0 | 0.0031 | 46.2 | 45.0 | 47.2 | 5.0 | 111.0 | 111.0 | 0.0003 |
| 9 | 80.0 | 85.0 | 81.6 | 4.0 | 186.0 | 186.0 | 0.0011 | 86.5 | 95.0 | 90.6 | 4.7 | 185.0 | 185.0 | 0.0006 |
| 10 | 106.3 | 106.0 | 109.4 | 3.2 | 223.0 | 223.0 | 0.0044 | 106.8 | 118.0 | 116.3 | 1.5 | 203.0 | 203.0 | 0.0062 |
| 11 | 57.3 | 57.0 | 58.5 | 2.6 | 136.0 | 136.0 | 0.0006 | 54.3 | 59.0 | 55.8 | 5.4 | 124.0 | 124.0 | 0.0024 |
| 12 | 44.4 | 50.0 | 44.9 | 10.1 | 115.0 | 115.0 | 0.0016 | 55.9 | 58.0 | 58.3 | 0.6 | 126.0 | 126.0 | 0.0000 |
| 13 | 77.1 | 79.0 | 80.5 | 2.0 | 162.0 | 162.0 | 0.0024 | 79.7 | 95.0 | 95.2 | 0.2 | 179.0 | 179.0 | 0.0016 |
| 14 | 56.2 | 57.0 | 56.6 | 0.7 | 144.0 | 144.0 | 0.0014 | 79.2 | 81.0 | 83.2 | 2.7 | 169.0 | 169.0 | 0.0013 |
| 15 | 50.1 | 55.0 | 50.5 | 8.1 | 132.0 | 132.0 | 0.0039 | 71.3 | 72.0 | 74.1 | 3.0 | 164.0 | 164.0 | 0.0006 |

CMR: cardiovascular magnetic resonance; FEM: finite element model; ESV: end-systolic volume; EDV: end-diastolic volume.

### 3.3. Baseline Characteristics and CMR Global Function

Our study consisted of 15 RHD patients and 15 controls. The RHD patients were well-matched with the controls for age, sex, height, weight, and BMI. There was a significant difference between the left ventricular volume of the RHD and the control group. The ejection fraction (EF) was lower in the RHD group compared to the control (LVEF: median = 51% vs. 57%, *p*-value = 0.036), while the LV end-diastolic volume and LV end-systolic volume significantly increased. The ECV and native T1, which are parameters that quantify the degree of diffuse myocardial fibrosis [45–48], were significantly different

between the groups (ECV: median = 34% vs. 28%, *p*-value = 0.001; native T1: 1290ms vs. 1213 ms, *p*-value = 0.001) (see Table 3).

**Table 3.** Baseline characteristics of study population and CMR global functions.

| Characteristics | RHD (n = 15) Median (IQR) | Controls (n = 15) Median (IQR) | *p*-Value |
|---|---|---|---|
| Sex | | | |
| Male n(%) | 6 (40) | 9 (60) | |
| Female n(%) | 9 (60) | 6 (40) | 0.237 |
| Age (years) | 38 (26–48) | 43 (36–54) | 0.309 |
| Height (cm) | 169 (158–175) | 172 (163–178) | 0.191 |
| Weight (kg) | 73 (64–95) | 83 (73–97) | 0.130 |
| BMI (kg/m$^2$) | 20 (23–32) | 29 (26–33) | 0.534 |
| LVEDV (mL) | 178 (136–243) | 136 (123–162) | 0.038 * |
| LVESV (mL) | 97 (77–119) | 57 (54–78) | 0.005 * |
| LVEF (%) | 51 (46–60) | 57 (53–61) | 0.036 * |
| RVEDV (mL) | 136 (109–185) | 164 (126–185) | 0.245 |
| RVESV (mL) | 78 (67–104) | 72 (61–95) | 0.309 |
| RVEF (%) | 42 (26–48) | 52 (46–56) | 0.001 * |
| Native T1 (ms) | 1290 (1248–1341) | 1213 (1194–1239) | 0.001 * |
| ECV (%) | 34 (30–39) | 28 (27–29) | 0.001 * |

RHD: rheumatic heart disease; IQR: interquartile range; BMI: body mass index; LVEDV: left ventricular end-diastolic volume; LVESV: left ventricular end-systolic volume; LVEF: left ventricular ejection fraction; RVEDV: right ventricular end-diastolic volume; RVESV: right ventricular end-systolic volume; RVEF: right ventricular ejection fraction; ECV: extracellular volume; * *p*-Value < 0.05.

*3.4. Estimation of Elastic Material Parameters*

As mentioned earlier in Section 2.6, we used Klotz's method as the existing standard procedure for verifying patient-specific end-diastolic pressure-volume relationships (ED-PVR) [36]. Figures 3 and 4 show the EDPVR curves produced from an RHD and control model in comparison to the Klotz curve. These figures show great agreement with the Klotz curve having $R^2$ = 0.987 for the RHD patient and $R^2$ = 0.958 for control subject. The final average end-systolic pressure in the LV was 0.04 kPa in the fifteen RHD patients and 0.02 kPa in the fifteen control subjects.

The estimation of the elastic material parameters based on the Klotz curve led to the elastic material properties in Tables 4 and 5. The range of fitting accuracy was found as $R^2$ = 0.953 to 1.000 and $R^2$ = 0.938 to 0.998 for the diseased and control groups, respectively. Table 6 shows the mean difference between the RHD and control material parameters and fiber angles. There was no significant difference in the RHD and control stiffness parameters in the right ventricles ($p$ > 0.05). Also, the initial diastolic filling stiffness parameter $A$ was significantly different in the left ventricle of both subjects. The direction-dependent parameters, $a_1 - a_6$, differed significantly between the RHD patients and the controls ($p$-values < 0.05). There was a significant difference in the left ventricular endocardial fiber angles of healthy and diseased hearts, but the LV epicardial fiber angles were not statistically significantly different ($p$ > 0.05).

In order to show the degree of changes in the direction-dependent material behavior, the anisotropy parameters $b_{ff}$ to $b_{sn}$ were computed by Equations (5a–f), which have a direct physical meaning, and their ratios were calculated as listed in Table 7. A significant difference can be seen in the anisotropic parameter ratios of the diseased and healthy subjects ($p$-value < 0.005). In particular, the axial stiffness ratio between the fiber and sheet directions, $b_{ff}/b_{ss}$, was noticeably elevated, whereas all other ratios were smaller for the RHD cases.

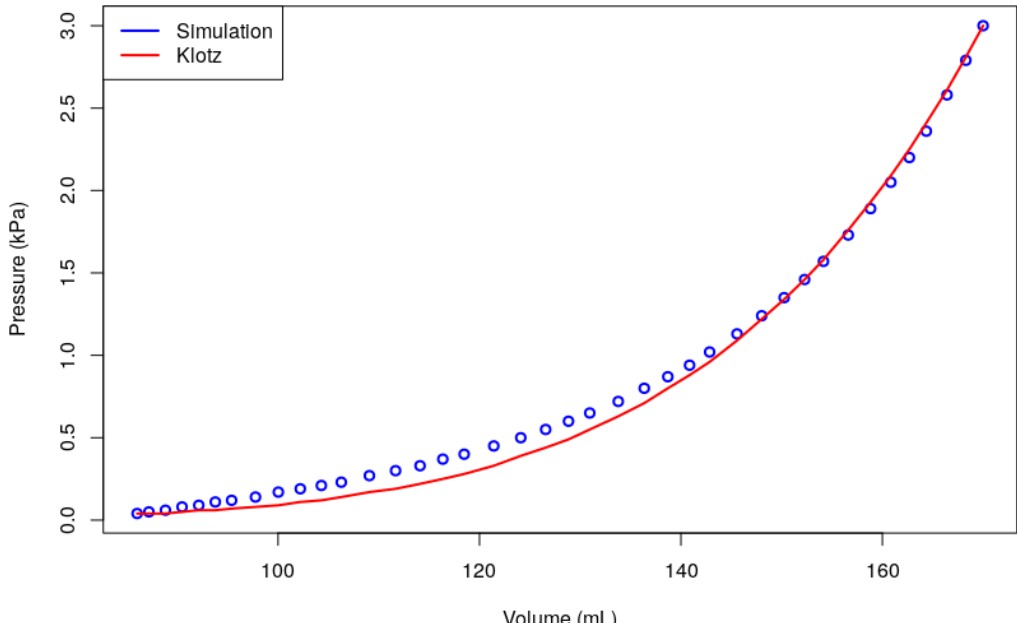

**Figure 3.** EDPVR curve predicted by the Usyk model (blue) and the method of Klotz et al. [36] (red) as a physiological benchmark for RHD Case 1.

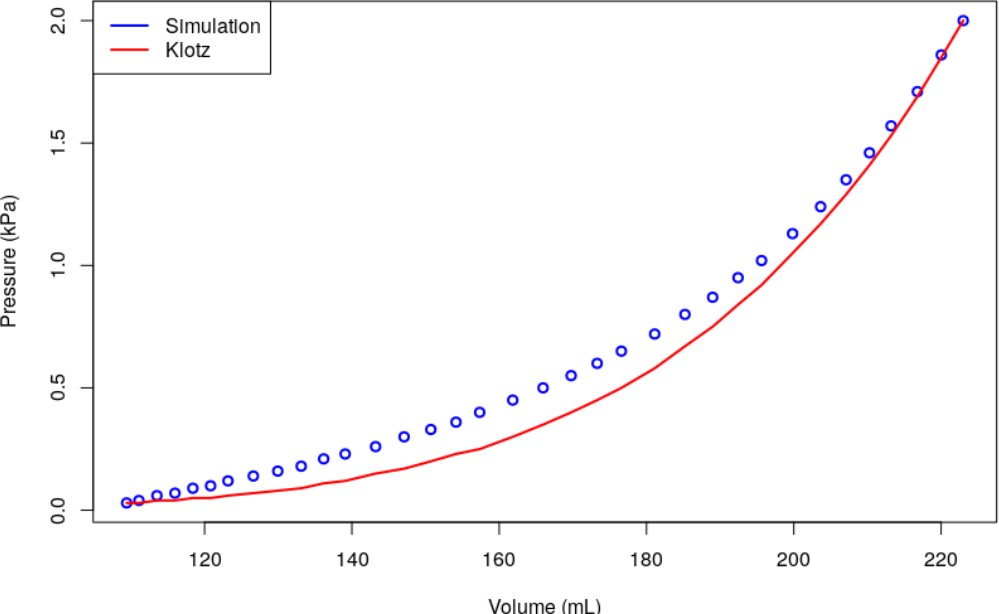

**Figure 4.** EDPVR curve predicted by the Usyk model (blue) and the method of Klotz et al. [36] (red) as a physiological benchmark for Control Case 1.

**Table 4.** Subject-specific in vivo elastic material parameter values of RHD patient myocardium with 3.0 kPa (22.50 mmHg) LVEDP and 1.0 kPa (7.50 mmHg) RVEDP.

| | Right Ventricle | | Left Ventricle | | | | | | |
|---|---|---|---|---|---|---|---|---|---|
| Subjects | A (kPa) | B (-) | A (kPa) | $a_1$ (-) | $a_2$ (-) | $a_3$ (-) | $a_4$ (-) | $a_5$ (-) | $a_6$ (-) |
| 1 | 0.10 | 1.64 | 0.09 | −6.04 | −5.43 | 10.22 | 15.31 | 13.37 | −6.58 |
| 2 | 0.11 | 1.35 | 0.10 | −6.11 | −5.39 | 10.26 | 15.55 | 14.32 | −6.40 |
| 3 | 0.10 | 1.07 | 0.11 | −6.48 | −6.10 | 14.32 | 20.41 | 17.53 | −6.32 |
| 4 | 0.07 | 1.02 | 0.14 | −6.55 | −5.74 | 12.55 | 16.79 | 15.53 | −6.19 |
| 5 | 0.34 | 1.29 | 0.31 | −6.06 | −5.19 | 9.65 | 13.64 | 12.94 | −6.16 |
| 6 | 0.35 | 1.67 | 0.32 | −6.88 | −6.49 | 16.87 | 25.80 | 20.27 | −7.30 |
| 7 | 0.10 | 1.33 | 0.11 | −6.48 | −6.54 | 17.02 | 24.65 | 19.88 | −6.50 |
| 8 | 0.10 | 1.17 | 0.11 | −6.62 | −6.70 | 17.73 | 26.12 | 20.32 | −6.70 |
| 9 | 0.11 | 1.70 | 0.11 | −6.43 | −6.23 | 15.33 | 22.57 | 19.35 | −6.39 |
| 10 | 0.11 | 1.56 | 0.10 | −6.46 | −6.24 | 15.14 | 22.59 | 19.00 | −6.58 |
| 11 | 0.11 | 1.40 | 0.10 | −6.20 | −5.52 | 11.27 | 16.50 | 14.71 | −6.38 |
| 12 | 0.73 | 1.30 | 0.66 | −6.71 | −6.80 | 19.10 | 28.03 | 22.85 | −6.42 |
| 13 | 0.24 | 1.56 | 0.22 | −6.89 | −7.07 | 18.69 | 27.53 | 22.24 | −6.63 |
| 14 | 0.12 | 1.52 | 0.11 | −6.57 | −6.80 | 17.77 | 26.62 | 22.34 | −6.50 |
| 15 | 0.10 | 1.12 | 0.11 | −6.57 | −6.47 | 16.30 | 23.65 | 19.84 | −6.48 |

**Table 5.** Subject-specific in vivo elastic material parameter values of healthy subjects' myocardium with 2.0 kPa (15.00 mmHg) LVEDP and 1.0 kPa (7.50 mmHg) RVEDP.

| | Right Ventricle | | Left Ventricle | | | | | | |
|---|---|---|---|---|---|---|---|---|---|
| Subjects | A (kPa) | B (-) | A (kPa) | $a_1$ (-) | $a_2$ (-) | $a_3$ (-) | $a_4$ (-) | $a_5$ (-) | $a_6$ (-) |
| 1 | 0.10 | 0.90 | 0.10 | −6.09 | −5.12 | 9.71 | 12.75 | 12.54 | −5.97 |
| 2 | 0.12 | 1.75 | 0.11 | −6.26 | −5.50 | 9.97 | 15.78 | 15.43 | −6.55 |
| 3 | 0.11 | 1.66 | 0.10 | −6.15 | −5.59 | 10.76 | 17.20 | 15.64 | −6.63 |
| 4 | 0.10 | 1.02 | 0.10 | −6.00 | −5.00 | 8.99 | 12.02 | 12.03 | −6.01 |
| 5 | 0.13 | 1.98 | 0.10 | −6.48 | −5.43 | 10.69 | 13.94 | 14.73 | −6.11 |
| 6 | 0.12 | 1.63 | 0.10 | −6.13 | −5.26 | 9.53 | 14.06 | 13.92 | −6.23 |
| 7 | 0.11 | 1.45 | 0.11 | −6.55 | −5.86 | 12.91 | 18.32 | 16.71 | −6.34 |
| 8 | 0.10 | 0.96 | 0.10 | −6.17 | −5.27 | 10.30 | 13.77 | 13.21 | −6.09 |
| 9 | 0.11 | 1.22 | 0.10 | −6.25 | −5.47 | 10.74 | 15.34 | 14.15 | −6.36 |
| 10 | 0.38 | 1.57 | 0.31 | −6.26 | −5.47 | 10.56 | 15.26 | 14.24 | −6.22 |
| 11 | 0.10 | 1.07 | 0.10 | −6.20 | −5.35 | 10.57 | 14.54 | 13.77 | −6.15 |
| 12 | 0.11 | 1.36 | 0.10 | −6.06 | −5.12 | 9.05 | 12.82 | 12.72 | −6.12 |
| 13 | 0.12 | 1.85 | 0.10 | −6.33 | −5.90 | 12.18 | 19.54 | 17.39 | −6.70 |
| 14 | 0.12 | 1.57 | 0.10 | −6.20 | −5.31 | 8.91 | 13.84 | 13.37 | −6.47 |
| 15 | 0.13 | 1.65 | 0.10 | −6.31 | −5.13 | 8.91 | 12.21 | 13.37 | −6.04 |

*3.5. Global Strain And Stress*

Tagged CMR provides global strain indicators for the longitudinal, circumferential, and radial myocardial strain component distributions, separately averaged over the entire LV and RV, respectively. The longitudinal direction is defined as pointing from the ventricle base to its apex (long axis) and in the transverse plane (short axis); the circumferential direction is defined as the tangent on the ventricle wall and the radial direction perpendicular to it. Further details can be found in the CVI24 user manual (https://www.circlecvi.com/docs/product-support/manuals/cvi42-user-manual-v5.6.pdf, accessed on 14 February 2023). In the FEM modeling software SESKA, the global strain measures were computed by means of the integration of the corresponding strain component distributions over the LV and RV wall domains. The simulated left ventricular global strain results in Table 8 revealed good agreement between the clinical results from the CMR and FE simulated results for the circumferential and radial directions. In the RHD group, the average global circumferential strain (GCS) for the in vivo measurement

was 22.2%, which compared very well with the FE simulations result of 24.2%. The global longitudinal strain (GLS) was 18.8% for the in vivo measurement and 9.1% for the FE model simulation. The global radial strain (GRS) was −14.7% for the in vivo measurement and −18.7% for the FE model simulation. The healthy control subjects showed similar comparisons: the average GCS for the in vivo measurement and FE simulations was 28.1 and 27.4%, respectively; the GLS for the in vivo measurement and FE simulations was 23.80 and 11.4%, respectively; the GRS for the in vivo measurement and FE simulations was −17.40 and −20.5%, respectively. The circumferential and radial strains in both groups were more consistent than the longitudinal strains. The observed difference between the healthy and diseased global strains were statistically significant, with $p < 0.05$.

**Table 6.** Statistical significance of changes regarding in vivo elastic material parameters and fiber orientation with an LVEDP of 3 kPa and 2 kPa for the RHD and control groups, respectively.

| Characteristics | RHD (n = 15) Median (IQR) | Controls (n = 15) Median (IQR) | *p*-Value |
|---|---|---|---|
| Right ventricle | | | |
| A (kPa) | 0.11 (0.10–0.24) | 0.12 (0.10–0.12) | 0.633 |
| B | 1.35 (1.17–1.56) | 1.57 (1.57–1.66) | 0.468 |
| Left ventricle | | | |
| A (kPa) | 0.11 (0.11–0.22) | 0.10 (0.10–0.10) | 0.003 * |
| $a_1$ (-) | −6.48 (−6.62–(−6.20)) | −6.20 (−6.31–(−6.13)) | 0.021 * |
| $a_2$ (-) | −6.24 (−6.70–(−5.52)) | −5.35 (−5.50–(−5.13)) | 0.001 * |
| $a_3$ (-) | 15.33 (11.27–17.73) | 10.30 (9.05–10.74) | 0.001 * |
| $a_4$ (-) | 22.59 (16.50–26.12) | 14.06 (12.82–15.78) | <0.001 * |
| $a_5$ (-) | 19.35 (14.71–20.32) | 13.92 (13.21–15.43) | 0.001 * |
| $a_6$ (-) | −6.48 (−6.58–(−6.38)) | −6.22 (−6.47–(−6.09)) | 0.015 * |
| $\theta_{endo}$ (°) | 69.63 (67.08–72.42) | 72.07 (71.30–74.45) | 0.006 * |
| $\theta_{epi}$ (°) | −58.45 (−59.8–(−57.1)) | −58.04 (−58.8–−56.9) | 0.395 |

IQR: interquartile range; LV: left ventricle; RV: right ventricle; A and B: stiffness parameters; $a_1$–$a_6$: anisotropy parameters; $\theta_{endo}$: fiber orientation at the endocardium; $\theta_{epi}$: fiber orientation at the epicardium; * *p*-Value < 0.05.

**Table 7.** Statistical significance of changes regarding the anisotropy nature of the myocardial tissue in the RHD and control groups using the ratios of the orientation-dependent parameters.

| Characteristics | RHD (n = 15) Median (IQR) | Controls (n = 15) Median (IQR) | *p*-Value |
|---|---|---|---|
| $b_{ff} : b_{ss}$ | 1.26 (1.12–1.36) | 0.96 (0.89–1.05) | <0.001 * |
| $b_{ff} : b_{nn}$ | 1.78 (1.73–2.10) | 2.07 (1.81–2.41) | 0.033 * |
| $b_{ss} : b_{nn}$ | 1.44 (1.27–1.88) | 2.10 (1.98–2.62) | <0.001 * |
| $b_{fs} : b_{sn}$ | 3.38 (3.23–3.75) | 3.68 (3.49–3.85) | 0.027 * |
| $b_{fs} : b_{fn}$ | 2.59 (2.43–3.08) | 3.51 (3.26–3.73) | <0.001 * |
| $b_{fn} : b_{sn}$ | 1.28 (1.22–1.34) | 1.07 (1.02–1.15) | <0.001 * |

IQR: interquartile range; RHD: rheumatic heart disease; * *p*-Value < 0.05.

The contour plots of myofiber stress $\sigma_f$ and strain $\epsilon_f$ are presented in Figure 5. The myofiber stress and strain in the deformed configuration are given by:

$$\sigma_f = \sigma : \mathbf{F}\mathbf{M}_f\mathbf{F}^T \tag{8}$$

$$\epsilon_f = \mathbf{e} : \mathbf{F}\mathbf{M}_f\mathbf{F}^T \tag{9}$$

where $\sigma$ denotes the Cauchy stress tensor, $\mathbf{e}$ the Almansi strain tensor, and $\mathbf{F}$ the deformation gradient tensor. These contour plots provide qualitative information about the

myofiber stress and strain distributions associated with the geometric position. We observed high stresses on the endocardial surface of the LV of RHD patients and higher strain on the septum of healthy controls. Quantitatively, the end-diastolic circumferential stress in the diseased group was 6.02 kPa, the longitudinal stress was 3.57 kPa, and the radial stress was 0.83 kPa. Likewise, the circumferential stress in the control group was 3.90 kPa; the longitudinal stress was 2.58 kPa; the radial stress was 0.82 kPa (see Table 9).

**Table 8.** Left ventricular clinical and computational global ventricular strains for RHD patients and controls.

|  | RHD (n = 15) | | Controls (n = 15) | |
|---|---|---|---|---|
|  | **CMR** | **FEM** | **CMR** | **FEM** |
| GLS (%) | 18.8 | 9.1 | 23.8 | 11.4 |
| GCS (%) | 22.2 | 24.2 | 28.1 | 27.4 |
| GRS (%) | −14.7 | −18.7 | −17.4 | −20.5 |

CMR: cardiovascular magnetic resonance; FEM: finite element model; GLS: global longitudinal strain; GCS: global circumferential strain; GRS: global radial strain.

**Table 9.** Left ventricular averaged myofiber stress $\sigma_f$ (kPa) for RHD patients and controls at the end of diastole.

|  | RHD (n = 15) Median (IQR) | Controls (n = 15) Median (IQR) |
|---|---|---|
| Longitudinal | 3.57 (3.16–3.80) | 2.58 (2.28–2.75) |
| Circumferential | 6.02 (5.09–6.30) | 3.90 (3.33–4.15) |
| Radial | 0.83 (0.77–0.93) | 0.82 (0.74–0.90) |

IQR: interquartile range.

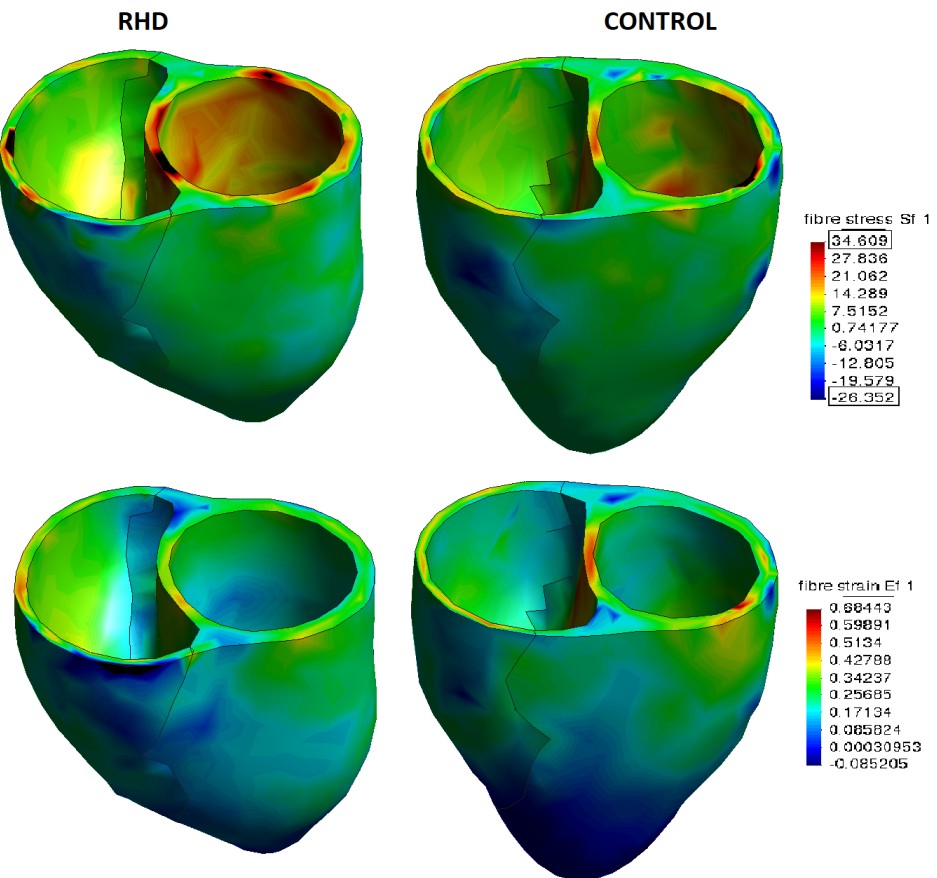

**Figure 5.** Myofiber stress $\sigma_f$ (**top**) and strain $\epsilon_f$ (**bottom**) of RHD patients and controls at the end of diastole.

### 3.6. Association between CMR and Simulated Parameters

Tests of association between the LVEF, the global strains, and the stiffness parameters are shown in Table 10. There was a significant association between the anisotropy coefficients parameters and the LVEF, GCS, GLS, and GRS; the *p*-values were <0.05; this shows that an increase in the myocardial tissue stiffness decreased the EF, GCS, and GLS and vice versa. The opposite is true for the GRS: an increase in the myocardial tissue stiffness increased the GRS. We further studied the correlation between the LVEF and the simulated global stresses and strains. As shown in Figure 6, a negative correlation was found between the GRS and the LVEF (R = −0.68, *p* < 0.001), while the GCS and GLS were positively correlated with the LVEF (GCS: R = 0.90, *p* < 0.001; GLS: R = 0.81, *p* < 0.001). Also, a positive correlation was found between the global radial stress and the LVEF (R = 0.76, *p* < 0.001), while the global circumferential stress was negatively correlated with the LVEF (R = −0.37, *p* = 0.042).

**Table 10.** Tests of association between LVEF, global strains, and elastic material parameters.

| | LVEF R (*p*-Value) | GCS R (*p*-Value) | GLS R (*p*-Value) | GRS R (*p*-Value) |
|---|---|---|---|---|
| A (kPa) | −0.584 (0.001 *) | −0.433 (0.017 *) | −0.400 (0.029 *) | 0.473 (0.008 *) |
| $b_{ff}$ (-) | −0.913 (<0.001 *) | −0.795 (<0.001 *) | −0.732 (<0.001 *) | 0.809 (<0.001 *) |
| $b_{ss}$ (-) | −0.925 (<0.001 *) | −0.781 (<0.001 *) | −0.729 (<0.001 *) | 0.797 (<0.001 *) |
| $b_{nn}$ (-) | −0.905 (<0.001 *) | −0.734 (<0.001 *) | −0.720 (<0.001 *) | 0.763 (<0.001 *) |
| $b_{fs}$ (-) | −0.924 (<0.001 *) | −0.788 (<0.001 *) | −0.735 (<0.001 *) | 0.804 (<0.001 *) |
| $b_{fn}$ (-) | −0.919 (<0.001 *) | −0.788 (<0.001 *) | −0.733 (<0.001 *) | 0.805 (<0.001 *) |
| $b_{sn}$ (-) | −0.927 (<0.001 *) | −0.774 (<0.001 *) | −0.731 (<0.001 *) | 0.794 (<0.001 *) |

LVEF: left ventricular ejection fraction; GCS: global circumferential strain; GLS: global longitudinal strain; GRS: global radial strain; * *p*-Value < 0.05.

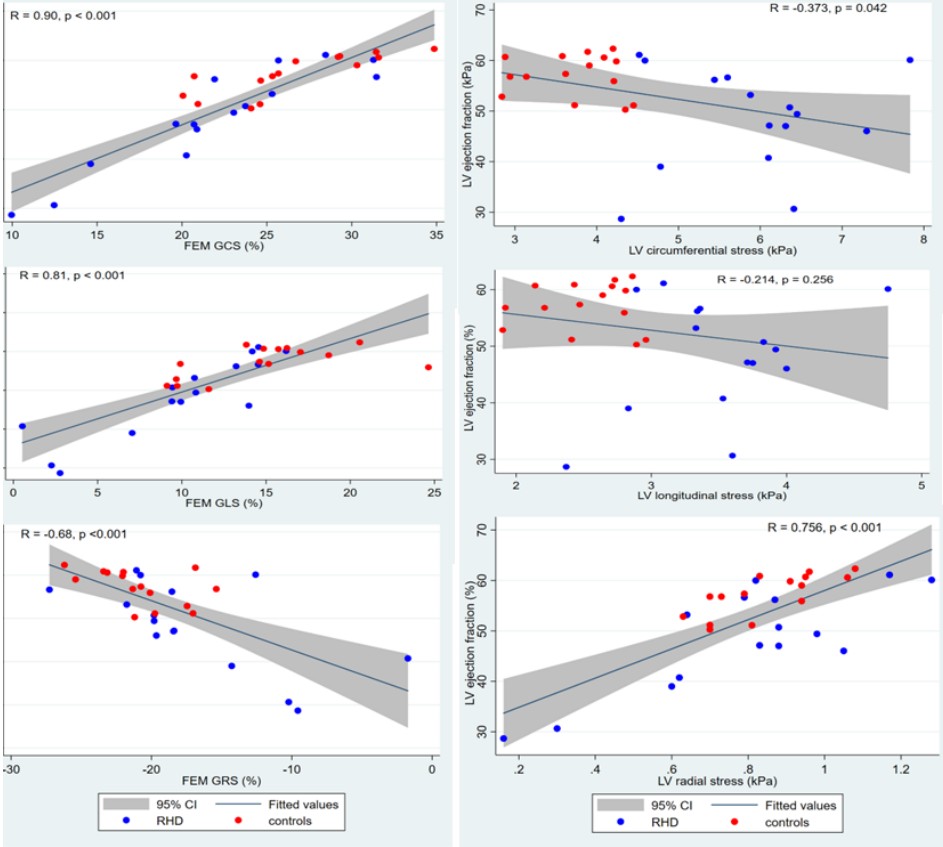

**Figure 6.** Correlation of the finite element model (FEM) global strains and stresses to the LVEF.

### 3.7. Sensitivity Analysis

Due to the unavailability of diastolic pressure, which required invasive measurement, we carried out a sensitivity analysis to examine the effect of a change in the LVEDP on the elastic material parameters. Using one of the subjects, the LVEDP was varied to 1.5, 2.0, 2.5, and 3.0 kPa (11.25, 15.00, 18.75, and 22.50 mmHg), while we fixed the RVEDP at 7.50 mmHg with the assumption that the RV was not impaired. The estimated values of the material parameters are depicted in Table 11. It was observed that an increase in the LVEDP increased the LV stiffness, but showed a decrease in the RV stiffness parameters. Similarly, the stiffness-orientation-dependent parameters, $a_i(i = 1, \ldots, 6)$, increased with an increased LVEDP. We also noted that the endocardial fiber angle decreased with an increase in the LVEDP. Meanwhile, when both the LVEDP and RVEDP were varied simultaneously (1.5 vs. 1.0, 2.0 vs. 1.0, 2.5 vs. 1.5, 3.0 vs. 2.0 kPa), we noted an increase in the stiffness of both ventricles. Another sensitivity study was carried out using the same LVEDP of 3.0 kPa for both the RHD and control groups. This was used to examine whether the observable difference in the elastic material parameters of the RHD and control groups (Table 6) was not due to differences in their LVEDP. The obtained results presented in Table 12 were similar to the results in Table 6. The elastic material parameters of the RHD group were significantly higher when compared to the healthy controls due to fibrosis in the deceased myocardium and not necessarily due to differences in the LVEDP.

**Table 11.** Parameter values estimated with different end-diastolic pressure for an RHD BiV.

| LVEDP (kPa) | Right Ventricle | | Left Ventricle | | | | | | | $\theta_{endo}$ (°) | $\theta_{epi}$ (°) |
|---|---|---|---|---|---|---|---|---|---|---|---|
| | A (kPa) | B | A (kPa) | $a_1$ | $a_2$ | $a_3$ | $a_4$ | $a_5$ | $a_6$ | | |
| 1.5 | 0.11 | 1.91 | 0.10 | −6.37 | −5.85 | 11.44 | 18.53 | 16.77 | −6.92 | 75.27 | −57.9 |
| 2.0 | 0.11 | 1.73 | 0.10 | −6.39 | −5.99 | 12.95 | 20.31 | 17.72 | −6.81 | 72.23 | −58.4 |
| 2.5 | 0.11 | 1.62 | 0.10 | −6.49 | −6.18 | 14.28 | 21.61 | 18.59 | −6.79 | 71.06 | −58.4 |
| 3.0 | 0.11 | 1.56 | 0.10 | −6.46 | −6.24 | 15.14 | 22.59 | 19.00 | −6.58 | 70.41 | −58.3 |

LV: left ventricle; RV: right ventricle; A and B: stiffness parameters; $a_1$–$a_6$: stiffness orientation-dependent parameters; $\theta_{endo}$: fiber orientation at the endocardium; $\theta_{epi}$: fiber orientation at the epicardium.

**Table 12.** Subject-specific in vivo elastic material parameters and fiber orientation with an LVEDP of 3.0 kPa (22.5 mmHg) for both the RHD and control groups.

| Characteristics | RHD (n = 15) Median (IQR) | Controls (n = 15) Median (IQR) | *p*-Value |
|---|---|---|---|
| Right ventricle | | | |
| A (kPa) | 0.11 (0.10–0.24) | 0.11 (0.10–0.11) | 0.310 |
| B | 1.35 (1.17–1.56) | 1.33 (0.94–1.52) | 0.351 |
| Left ventricle | | | |
| A (kPa) | 0.11 (0.11–0.22) | 0.10 (0.10–0.11) | 0.013 * |
| $a_1$ (-) | −6.48 (−6.62–(−6.20)) | −6.31 (−6.39–(−6.26)) | 0.141 |
| $a_2$ (-) | −6.24 (−6.70–(−5.52)) | −5.59 (−5.86–(−5.38)) | 0.021 * |
| $a_3$ (-) | 15.33 (11.27–17.73) | 11.83 (10.59–13.17) | 0.027 * |
| $a_4$ (-) | 22.59 (16.50–26.12) | 16.14 (14.57–19.66) | 0.008 * |
| $a_5$ (-) | 19.35 (14.71–20.32) | 15.20 (13.84–17.23) | 0.021 * |
| $a_6$ (-) | −6.48 (−6.58–(−6.38)) | −6.21 (−6.52–(−6.13)) | 0.029 * |
| $\theta_{endo}$ (°) | 69.63 (67.08–72.42) | 70.91 (70.51–72.37) | 0.093 |
| $\theta_{epi}$ (°) | −58.45 (−59.8–(−57.1)) | −57.49 (−57.8–(−56.4)) | 0.085 |

IQR: interquartile range; LV: left ventricle; RV: right ventricle; A and B: stiffness parameters; $a_1$–$a_6$: stiffness-orientation-dependent parameters; $\theta_{endo}$: fiber orientation at the endocardium; $\theta_{epi}$: fiber orientation at the epicardium; * *p*-Value < 0.05.

Lastly, it is shown that the material behavior of myocardial muscle tissue is sensitive to the degree of incompressible material behavior enforced by the penalty parameter $A_{comp}$;

see, e.g., [49]. Therefore, a sensitivity analysis was performed with regard to the fitting accuracy of the model in relation to the Klotz EDPVR to determine the most-suitable value for the penalty parameter within the range of [50; 500] kPa. $A_{comp}$ = 100 kPa produced the closest agreement with RMSE = 0.446 and $R^2 = 0.999$ (see Figure 7). All fitted parameters for a respective $A_{comp}$ value are listed in Table 13. It can be seen that the base stiffness parameter $A$ controlling the material response in the linear stress–strain regime was not sensitive to the penalty parameter in the given range. The stiffness parameter controlling the nonlinear regime, which is parameter $B$ for the RV and $a_1$–$a_6$ for the LV (where $B = 1.0$ was fixed), exhibited a relatively small sensitivity in the vicinity of $A_{comp} = 100$ kPa. There was a slight tendency for the degree of anisotropy to increase with $A_{comp}$ as shown by the ratios of the orientation-dependent material parameters listed in Table 14. As can be expected, the simultaneous increase of the volumetric stiffness and degree of anisotropy was accompanied by a gradual increase of the stored strain energy.

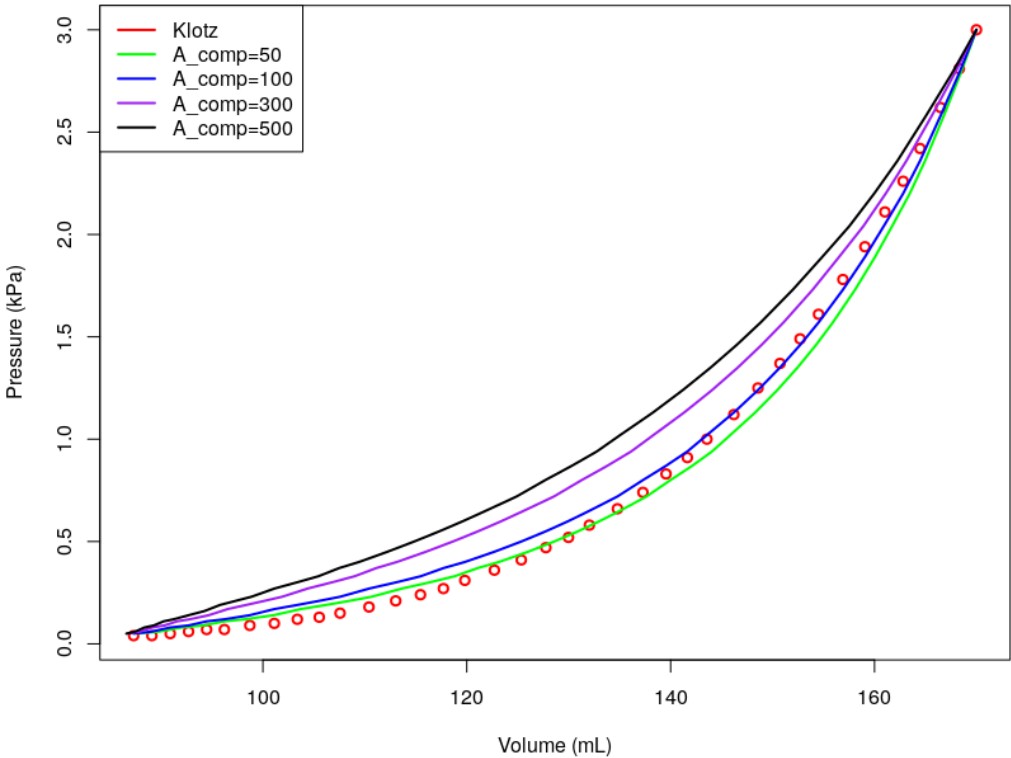

**Figure 7.** Sensitivity analysis of the penalty parameter $A_{comp}$ in relation to the Klotz EDPVR using RHD Case 1, $A_{comp}$ = 50 kPa ($RMSE$ = $0.552, R^2$ = $0.997$), $A_{comp}$ = 100 kPa(RMSE = 0.446, $R^2 = 0.999$), $A_{comp}$ = 300 kPa ($RMSE$ = $1.205$, $R^2 = 0.992$), $A_{comp} = 500$ kPa ($RMSE = 1.718$, $R^2 = 0.982$).

**Table 13.** Comparison of different $A_{comp}$ values with optimized material parameters using RHD Case 1.

| $A_{comp}$ (kPa) | Right Ventricle | | Left Ventricle | | | | | | | $\theta_{endo}$ (º) | $\theta_{epi}$ (º) |
|---|---|---|---|---|---|---|---|---|---|---|---|
| | A (kPa) | B (-) | A (kPa) | $a_1$ (-) | $a_2$ (-) | $a_3$ (-) | $a_4$ (-) | $a_5$ (-) | $a_6$ (-) | | |
| 50 | 0.11 | 1.74 | 0.11 | −6.60 | −6.39 | 15.90 | 23.14 | 19.91 | −6.42 | 71.44 | −58.9 |
| 100 | 0.11 | 1.70 | 0.11 | −6.57 | −6.37 | 15.68 | 23.08 | 19.78 | −6.54 | 71.57 | −59.5 |
| 300 | 0.11 | 1.42 | 0.11 | −6.82 | −6.45 | 15.22 | 22.33 | 18.96 | −6.87 | 71.26 | −59.3 |
| 500 | 0.11 | 1.20 | 0.11 | −7.16 | −6.56 | 14.59 | 21.23 | 18.47 | −7.12 | 71.19 | −58.1 |

A and B: stiffness parameters; $a_1$–$a_6$: stiffness-orientation-dependent parameters; $\theta_{endo}$: fiber orientation at the endocardium; $\theta_{epi}$: fiber orientation at the epicardium.

**Table 14.** Comparison of different $A_{comp}$ values with the ratios of the orientation-dependent parameters and strain energy for the LV using RHD Case 1.

| $A_{comp}$ (kPa) | Left Ventricle | | | | | | $e$ (kJoule) |
|---|---|---|---|---|---|---|---|
| | $b_{ff} : b_{ss}$ | $b_{ff} : b_{nn}$ | $b_{ss} : b_{nn}$ | $b_{fs} : b_{fn}$ | $b_{fs} : b_{sn}$ | $b_{fn} : b_{sn}$ | |
| 50 | 1.22 | 1.74 | 1.43 | 2.57 | 3.19 | 1.24 | $3.94 \times 10^5$ |
| 100 | 1.23 | 1.81 | 1.47 | 2.59 | 3.24 | 1.25 | $4.06 \times 10^5$ |
| 300 | 1.24 | 1.86 | 1.50 | 2.67 | 3.42 | 1.28 | $4.24 \times 10^5$ |
| 500 | 1.18 | 1.88 | 1.59 | 2.81 | 3.50 | 1.24 | $4.30 \times 10^5$ |

$b_{ff}, b_{ss}, b_{nn}, b_{fs}, b_{sn}$: stiffness parameters (Equation (5)); $e$: stored strain energy (Equation (3)).

## 4. Discussion

In this study, a quantitative analysis was performed to examine the difference between the elastic material parameters of 15 healthy controls and 15 RHD patients. To non-invasively obtain a parameter estimation of the constitutive model from the routinely used clinical CMR data, a combination of the FE simulations and LVM optimization schemes [28] was used. Using the end-systolic phase of the extracted tagged CMR images as a patient-specific anatomical computer model, the material parameters obtained reproduced the clinically measured EDV for the LV and RV within the physiological EDP [30,50,51]. This study included 30 subject-specific FE models, personalized with in vivo geometry and volumes measured with CMR. Comparing RHD patients and healthy subjects, there was a significant difference in the data-derived metrics between the two groups. There was a noticeable increase in both the end-diastolic and end-systolic cavity volumes in the RHD hearts, as widely reported in the literature [52,53]. The ejection fraction was considerably lower in the RHD group, thereby indicating impaired tissue contractility and diastolic filling.

From computational parameter optimization, it was found that the RHD subjects yielded material parameters of greater magnitude (Table 6) than the healthy subjects. The stiffness parameter *A*, which controls the baseline relation between stress and strain for smaller strains at the initial diastolic filling, was not significantly different in both the RHD and healthy groups. Meanwhile, parameter *B*, which controls the nonlinear relation between stress and strain for larger strains when the collagen fiber stiffness is mobilized, was significantly higher in the RHD group compared to the healthy group. Our result is consistent with clinical and modeling studies [54,55], which report that increased myocardial stiffness is generally associated with inflammatory heart disease. The reason is that a structural change in the collagen content in the extracellular matrix [56,57] and the presence of fibrosis are associated with increased stiffness [58]. The structure and function of the myocardium are affected by the excessive deposition of extracellular matrix proteins that are rich in collagen [59], which characterize myocardial fibrosis. Arrhythmias, enlarged heart chambers, and deteriorated systolic and diastolic function are also associated with it [60,61].

The LVEF is regarded as a significant measure for assessing LV function. In line with earlier studies [62], we discovered that the LV myocardial strains obtained from the simulations were significantly correlated with the LVEF. This is plausible, because the LVEF measures the fractional change in three-dimensional volume and the strain determines the fractional volume increase during diastolic filling [63]. Also, our analysis showed that, as the LVEF reduced, there was an increase in the stiffness and its orientation-dependent parameters. Furthermore, the LV myocardial circumferential and longitudinal stresses obtained from the simulations were negatively correlated with the LVEF. In other words, as the ejection fraction decreased, there was an increase in the myocardial stress level. The decrease of ventricular compliance (increase of stiffness) was clearly consistent with the reduction of the strain and the increase of the stress from a mechanics point of view, also confirmed by [64]. The observed relationship between the LVEF and the simulated my-

ocardial strains and stresses showed that the FEM model captured the heart's physiological behavior, as expected.

The directional material behavior was significantly altered in the RHD group, being more pronounced in terms of the dominant axial stiffness ratio $b_{ff}/b_{ss}$, and diminished in all other parameter ratios. Furthermore, the GCS, GLS, and GRS showed a significantly strong negative association with the anisotropy coefficients. This implies that the exhibited change in the anisotropic nature of a material reduced its ventricular compliance. This result was confirmed by [65], who found that the deformation was reduced as the anisotropic ratio increased in a study of BiV models using the anisotropic Holzapfel–Ogden material model.

The results from the comparison of the in vivo and model-predicted strains showed that the global circumferential and radial strains matched very well, but the longitudinal strains matched with less accuracy. In both groups, the model-predicted strains were lower than the in vivo measurements from using CMR. This result compares with the finding in [66] in that the FE strains were lower than the in vivo measurements. The discrepancy between the clinical and FEM strains may result from the absence of data on the true distribution of the patient-specific myofiber orientation angle, which affects the circumferential–longitudinal compliance ratio. Also, the mismatch between the longitudinal strain results may be linked to the fact that CMR short axis view (SAX) and long axis view (LAX) had different timing for the same cardiac cycle phase; both the circumferential and radial strain were obtained from the SAX, and the longitudinal strains were obtained from the LAX. Meanwhile, the FE model used the same time for all three directions. These observed smaller strains in the diseased hearts as compared to the healthy heart agreed with the previous findings [67–69], which suggests global LV impairment.

One benefit of computer modeling is the precise estimation of stress within complex mechanical problems. This is particularly important in terms of cardiac mechanics, because it is believed that variations in ventricular wall stress are what cause pathological remodeling [70–72]. In the contour plot shown in Figure 5, the LV of the RHD patient experienced high tensile stresses at the endocardium and the epicardium experienced compression. Also, we could observe variations in the healthy subject and RHD patient myofiber stress and strain distribution over the entire myocardium. In the diseased group, there was higher myocardial diastolic stress in the circumferential and longitudinal directions, but lower stress in the radial direction. This was to be expected, because the diastolic filling phase was determined by the circumferential and longitudinal areas' expansion and elongation, respectively. Because the myocardial tissue of the diseased heart has become less compliant due to fibrosis, it is unavoidable that increased stress is seen in both directions due to the effort required to allow for blood filling during the diastolic phase. We refrained from comparing the myocardial stress results of the healthy patients to those of the RHD subjects due to the varied assumed EDP employed in the calibration exercise for both groups. Despite this, we demonstrated that the increased elastic material parameters observed in the diseased group compared to the healthy participants were not attributable to increased pressure, but rather to the presence of fibrosis in the myocardium.

The subject-specific EDP was not available due to invasive measurement and technical limitations [8,29,30,55,73]. Therefore, a sensitivity study was performed to examine the effects of the change in the LVEDP on the elastic material parameters. Using one of the subjects, we discovered that the change in the LVEDP caused an increase in the elastic material parameters. This result resonates with the experimental study reporting an increase in the left ventricular stiffness due to increased EDP during a handgrip exercise [74]. Similarly, the stiffness-orientation-dependent parameters increased with an increased LVEDP. Additionally, we observed that the fiber angle at the endocardium decreased as the EDP increased, but increased the epicardial fiber angle. In the second sensitivity analysis, to ensure that the observed considerable difference between diseased and healthy myocardium was not solely due to differences in the LVEDP, we applied the same pressure to both groups. The results obtained are comparable to when different pressures were imposed; the RHD patients had increased elastic material parameters when compared to

healthy controls, which could be attributed to fibrosis in the deceased myocardium and not necessarily due to differences in the LVEDP in both groups.

*Model Limitations*

In this study, we used biventricular models to reduce the error in parameter estimate accuracy as compared to solely considering the LV, because the Klotz curve isonly directly applicable to the left ventricle. In the RV, we simply calibrated for the two stiffness parameters, leaving the six anisotropy parameters $a_1-a_6$ as originally proposed by [23]. The disparities between the clinical strains and those predicted by the model were notably large (Table 8). Among others, the calibration of the elastic material parameters depends on the EDP and fiber orientation. We only calibrated the LV fiber angles in this study, but the RV fiber angles were fixed to a physiologically meaningful value. Heart catheterization is the gold standard for acquiring the LVEDP, and diffusion tensor magnetic resonance imaging data can be used to approximate patient-specific fiber orientation. Furthermore, the model-predicted strains may be considerably impacted by atrial contraction, which was not considered in this study. Our computational model lacks physiological features like the diaphragm, pericardium, and atria. Our material model of the heart is constrained in that it does not take into account regional tissue heterogeneity or microstructural mechanics.

## 5. Conclusions

In this study, subject-specific cardiac models for the biventricular human heart in both healthy and diseased conditions were introduced. We provided a method for determining the mechanical parameters of the myocardium in beating hearts from computer simulations. In this study, inverse optimization was used to predict subject-specific in vivo elastic material parameters of human biventricular human models in healthy and rheumatic heart disease patients' myocardium. The elastic diastolic mechanics were simulated using CMR-based anatomically accurate subject-specific models of thirty human biventricles (BiVs), which utilized a orthotropic constitutive law. The simulated results showed a moderate agreement of global strains with the in vivo global strains. The inclusion of CMR strains as target values in the model calibration and making use of subject-specific fiber orientation would significantly improve the model's accuracy. Importantly, the anatomical models showed that the diseased hearts had increased myocardial tissue stiffness when compared to the healthy hearts. The expected stress and strain attributes in healthy and diseased hearts were qualitatively captured. Notably, we discovered a significant difference between the level of anisotropy of the RHD and healthy subjects using the ratios of the orientation-dependent parameters. The diseased heart undergoes changes in the micro-structural tissue composition due to fibrosis. These results may be useful for future computational research on heart failure treatment.

**Author Contributions:** The study's idea and design were overseen by M.A.F., S.S. and N.A.B.N. The computational modeling and statistical analyses were performed by M.A.F. The initial draft of the work was written by M.A.F. All authors helped to revise the work. All authors have read and agreed to the published version of the manuscript.

**Funding:** The research work received support from the South African DST-NRF Centre of Excellence in Epidemiological Modelling and Analysis (SACEMA) and the National Research Foundation (NRF) of South Africa (UID 142877). The opinions and conclusions expressed in the research are solely those of the authors and should not be attributed to SACEMA or NRF. Also, N.A.B.N. acknowledges funding from the South African Medical Research Council, the National Research Foundation and, the Lily and Ernst Hausmann Trust.

**Data Availability Statement:** The data that support the findings of this study are available from the corresponding author upon request.

**Conflicts of Interest:** The authors declare no conflict of interest.

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
