# Peer review of "Model-Based Assessment of Elastic Material Parameters in Rheumatic Heart Disease Patients and Healthy Subjects"

_mca, doi:10.3390/mca28060106_

Round 1

Reviewer 1 Report

Comments and Suggestions for Authors

Having read the manuscript with the ID: mca-2251945, for the Journal MCA (ISSN 2297-8747), I have the following major comments:

1) Reference configuration or unloaded shape of the heart should be estimated and used for such analysis. If the ESV is taken as a reference configuration the results would be different and incorrect, and this should be clarified in the manuscript.

2) From the results of CMR or documents of the patients, the biomarkers or at least the important ones should be presented for whole patients. The biomarkers that represent the severity of Rheumatic Heart Disease or Heart Failure are very important to be reported to the readers.

3) It has been mentioned that SESKA software has been used for calibration, but it has not been mentioned how different the calibrated model became, eventually. Also, it is not clear if you have one model for all patients and if so, how it would be a patient-specific study, as several times mentioned in the manuscript.

4) In section 3.5, where the association between CMR and simulated parameters is presented, needs more clarification on how the parameters in the simulation were calculated. For example, in CMR the definition for GLS is different from what we have in FEA and should be assigned to the model as well as its calculation for that matter. In the manuscript, some images or formulations should be presented to define what exactly the cMRI reported, and how they were calculated in FEA.

4) There are some important biomarkers in cardiology that can simply be defined and extracted from FEA simulation and the authors should train and validate the model for those biomarkers. For example, TAPSE can be calculated from simulation and compared with the patients’ reports. TAPSE is one of the most important biomarkers in Rheumatic Heart Disease and Heart Failure, and fortunately, TAPSE is available in any echo o CMR report. 

Reviewer 2 Report

Comments and Suggestions for Authors

Familusi et al. study the elastic material parameters of cardiac tissue in  health and disease. The manuscript is well written and structured and it discusses an important and interesting topic. However, some important parts in the methods section - that would enable other researchers to repeat this appraoch - are missing. Further, the choice of boundary conditions and initial values are not physiological. Before being able to assess the quality of results and conclusions coming from the computer simulations in this paper, the authors have to address some major concerns:          

1) "To simulate the mechanics of the diastolic filling phase of the heart it is necessary to apply appropriate boundary conditions". While there is a twisting motion at the apex in diastolic filling there is also a shift of the base in vertical direction upward. This is prohibited by the fixed base. More physiological would thus be spring like boundary conditions at the base and normal spring type boundary conditions at the epicardium - these are probably the elastic boundary conditions mentioned at the end of page 3. Further fixing the displacement of one point at the apex in x and y direction is also not physiological as the apex moves slightly in the pericardial sac. In previous papers the spring type BC at the base and normal spring type BCs at the epicardium where sufficient to get a converging algorithm, e.g., Hirschvogel et al. 2016 DOI: 10.1002/cnm.2842)

2) Optimization procedure: the core of the present paper seems to be the optimization of material parameters. However, in the methods section only a few lines are dedicated to this procedure in Section 2.5. The only citation mentioned is a general paper to the Levenberg-Marquardt algorithm and even there the authors state that they slightly modified the approach. The authors should elaborate on their technique to estimate material parameters s.t. interested readers could repeat the procedure with their own models. Further, no convergence criteria are given. Also, most likeley, the algorithm required a considerable number of forward simulations as it seems that all parameters given in Table 1 were used for the fitting. The authors should give the number of forward simulations needed as well as computational times to assess the efficiency of the method.

3) Initial parameters: where do the starting values come from? In Table 1 readers could get the impression that these initial values come from the work of Usyk et al. 2000. However as far as I see this is not the case. Further, negative values for some of the parameters, especially a1 and a2, are highly unusual for cardiac modeling. I could not find another work using negative values for the Usyk material for cardiac mechanics. This can likely lead to highly non-convex strain-energy functions that will have caveats in stability or could even give inaccurate results, see e.g., Section 2.4 in [Sun, Sacks, 2005, DOI: 10.1007/s10237-005-0075-x] or [Merodio, Ogden, 2006 DOI: 10.1016/j.ijengsci.2005.01.001]. Initial parameters should definitely come from the literature - e.g., Usyk et al. 2000 or more recent works - which are based on experimental data.

4) Authors should show loading curves between ESV and EDV. Further, authors should show comparisons to measured or estimated EDPVR (e.g. [Klotz et al 2007 DOI: 10.1152/ajpheart.01240.2005]) as this is the state-of-the-art in previous papers dealing with the parameter estimation of passive myocardial material. Amongst many others this includes the works of
[Sack et al, 2018 DOI: 10.3389/fphys.2018.00539],
[Marx et al 2022, DOI: 10.1016/j.jcp.2022.111266],
[Hadjicharalambous et al. 2015, DOI: 10.1007/s10237-014-0638-9].

5) The choice of A_comp is not given; sensitivity analysis should include the sensitivity to A_comp

6) The chosen mesh is very coarse (3500 linear tetrahedral elements) which is very likely to give inaccurate results due to locking effects. Thus the sensitivity analysis should also include a comparison of the solution with much finer grids.                                                                         

Minor:                                                                          

- typo in Section title of 2.1                                                  

- p 2, l 75; would be nice to have some reference to SESKA                    

- p 3, l 80; was there a pressure assumed in the end-systolic geometry or was this 0 kPa?

- p 3, Eq 2: Q around this equation should probably be Q_m 

Reviewer 3 Report

Comments and Suggestions for Authors

The manuscript by Familusi et al. presents a modeling procedure to simulates the cardiac biventricular function by properly calibrate the passive mechanical parameters for both the left and right cardiac ventricles to match the in vivo clinical end diastolic volume data very well. Specifically, to do it, they use an optimization procedure based on Levenberg-Marquardt method to estimate the myocardial constitutive material parameters. The analysis is carried out on 30 biventricular human geometries (15 healthy and 15 affected by rheumatic heart disease, RHD) derived from cardiac magnetic resonance-tagged cine images. A quantitative analysis was also performed to examine the difference between the elastic material parameters of healthy and RHD patients, showing that the RHD subjects yielded material parameters of greater magnitude with respect to the healthy ones.

The manuscript is sufficiently organized and presented, missing however of few relevant mathematical details. Although the study is interesting, there are major weaknesses and minor adjustments to be implemented for a journal publication.

General consideration:

In all the manuscript, it should be much more appropriate to refer at the term biventricular model (as properly used in line 70) instead of heart model. Indeed, what is considered in this study is a biventricular cardiac model and not a heart model, which is usually referred to the whole heart in the current cardiac modeling research community. As an example, in the Abstract (lines 3-4): A three-dimensional finite element model of the human heart was created => A three-dimensional finite element model of the human ventricles was created.

Note: see the References at the end of this Review, please.

Minors:

1.       Abstract, line 2: This study aimed => This study aims. (Simple present tense is usually preferred in the Abstract)

2.       Method, line 73: Add a reference or, at least, a footnote-link for the used meshing software GiD (https://www.gidsimulation.com/).

3.       Method, 75-76: add a refence and or a footnote-link to SESKA. Consider also to include some useful information concerning this FEM library (e.g. C++ or python based library, … developed by ...).

4.       Figure 1(c): Is that referred to the geometrical representation of the segmented configuration? Please, consider to also add a Figure 1(d) showing the computational mesh to fully clarify the pipeline procedure to build the computational mesh model.

5.       Method, line 82: Please, consider using mmHg, instead of kPa, as a unit of measure for the pressure because is the one most used in clinal practice.

6.       Method, line 83-84: Please, add also in the text that you are using a Rule-Based method by Wong et al. (also adding a reference) for prescribing the fiber architecture, as also specified in the caption of Figure 2.

7.       Please also add in the limitation of your work that you are prescribing the same fiber orientations in both ventricles, instead of using different fiber orientation for the LV and RV (like in [1,2]).

8.       Equation 1, line 89: please, use scientific notation for exp. Moreover, define all the other mathematical quantities embedded in the equation but not mentioned in the text.

9.       Method, line 94-95: please use for the fiber, sheet and sheet-normal directions the variable f, s and n, respectively. See for example [1].

10.   Method, lines 105-106: Is the epicardial BC a Robin boundary condition that mimic the effect of pericardial sac on the epicardium (like in [3,4])?

11.   Line 107: If not specifically related, use cited article instead of considering solely a PhD thesis [15], please.

12.   Line 111: myocardial wall => endocardial wall.

13.   Figure 2(d): Please, consider adding a figure to visualize the biventricular fiber architecture using, for example, streamlines or arrows. Add a citation to Wong a Kuhl article, please.

14.   Line 117: BV => BiV (to correct in all the manuscript).

15.   Table 1: Use degrees as unit of measure for theta angles, please.

16.   Lines 132-133: Clarify this sentence, please: “Clinical data was used to create the benchmark upon which calibration was done”.

17.   Lines 133-134: Clarify this sentence, using mathematical details (see also Point 4 of Majors), please: “The FEM modeling is based on a total Lagrangian approach where the undeformed configuration of the heart is used”.

18.   Lines 134-136: Please clarify the sentence described in these lines (134-136).

19.   Figure 3: Please, put in evidence the relevance of Figure 3.

20.   Lines 145-147: Please, add standard references for “Shapiro-Wilk test” and “Mann-Whitney U test”.

21.   Lines 150-151: If possible, add a footnote-link or a reference to “Stata software version 15 (College Station, TX, USA)”.

22.   Lines 156-157: If I correctly understood the LVEF and RVEF are "measured" based upon ESV and EDV coming from cine-CMR. Is it that correct?

23.   Line 158: Define, also in the text, the ECV acronym the first time it is used. Is this parameter related to the fibrosis or not? How is it measured?

24.   Table 3: Please define the IQR acronym the first time it is used.

25.   Line 184: Please define the GRS acronym the first time it is used.

26.   Line 193: How did you numerically (and experimentally) measured the strains?

27.   Lines 194-195: Please, clarify the differences, also mathematically defining them, among GCS, GLS and GRS.

28.   Lines 194-195: In the RHD group, the GCS coming from FE simulations (21.92%) mismatched to the reported one in Table 7 (17.8%). The same goes for GCS and GRS. Please also control the values for the healthy subjects.

29.   Line 210: How the myofiber stress are numerically computed?

30.   Figure 4: The Legend values is not clearly visible.

31.   Line 280: Miss pointer to ref [25].

32.   Line 283: “In other words, as the ejection fraction decreases there is an increase in the myocardial stress level.” This is in accordance with [5].

Majors:

1.       The authors should consider extending the literature review in the Introduction to account for recent advancements in the last years in the cardiac modeling. Only very few works are cited from 2016 to 2023. See, e.g., the reference list at the end of this review, please. Specifically, citations are required, for example, at the end of these sentences:

a.       Line 28: …(CVD).

b.       Line 29: …information [4].

c.       Line 31: …clinical data.

d.       Line 40-41: …to the clinic.

2.       Method, line 77 (3500 linear tetrahedral): please, also add the average mesh size for the meshes. 3500 linear tetrahedral seems to me a very coarse resolution that is not fine enough to correctly solve the cardiac mechanics problem. Standard value for the average mesh size is 3mm.

3.       Method, line 80. Is not clear to me how did you recover the reference stress free configuration. Indeed, you are acquiring cardiac geometries from in vivo medical images through imaging techniques. These geometries are in principle not stress free, due to the blood pressure acting on the endocardia. Therefore, you need to compute the unloaded (i.e., stress-free) configuration (also named reference configuration) to which the mechanical model is then applied and solved.

4.       Section 2.3: Please, add more details about the type of PDE equation (i.e. the momentum conservation equation) you are solving for the mechanics problem. Fully describe the mathematical treatment of the mechanical boundary conditions, only mentioned in Section 2.4, within the mathematical PDE system.

5.       Section 2.4, lines 102-103: The almost fixed ventricular base is a great modeling limitation of this work. More sophisticated BC should be considered: see [Piersanti et al., Augustin et al.]. Please, add this limitation to Section 4.1.

6.       Method, lines 107-108 (also related to the previous Point 5): With proper boundary condition in the biventricular base and epicardial surface you should obtain physiological base to apex displacement. Hence, there is no need to fix the ventricular base. This is not physiological.

7.       Section 2.5: Extend the mathematical details about the Levenberg-Marquardt method specifically related in this inverse modeling optimization procedure. Is not fully clear to me how this method is used to target subject specific hemodynamical quantities.

List of References:

[1] R. Doste et al. “A rule-based method to model myocardial fiber orientation in cardiac biventricular geometries with outflow tracts". In: International Journal for Numerical Methods in Biomedical Engineering 35.4 (2019).

[2] R. Piersanti et al. “Modeling cardiac muscle fibers in ventricular and atrial electrophysiology simulations”. Computer Methods in Applied Mechanics and Engineering 373 (2021).

[3] Pfaller et al. "The importance of the pericardium for cardiac biomechanics: from physiology to computational modeling". Biomechanics and Modeling in Mechanobiology 18.2 (2019).

[4] M. Strocchi et al. “Simulating ventricular systolic motion in a four-chamber heart model with spatially varying robin boundary conditions to model the effect of the pericardium". In: Journal of Biomechanics 101 (2020).

[5] Piersanti et al. "3D–0D closed-loop model for the simulation of cardiac biventricular electromechanics." Computer Methods in Applied Mechanics and Engineering 391 (2022).

[6] Augustin et al. "A computationally efficient physiologically comprehensive 3D–0D closed-loop model of the heart and circulation." Computer methods in applied mechanics and engineering 386 (2021).

[7] Gerach et al. "Electro-Mechanical Whole-Heart Digital Twins: A Fully Coupled Multi-Physics Approach". Mathematics 9.11 (2021).

[8] Regazzoni et al. "A cardiac electromechanical model coupled with a lumped-parameter model for closed-loop blood circulation." Journal of Computational Physics 457 (2022).

[9] Guan et al. Effect of myofibre architecture on ventricular pump function by using a neonatal porcine heart model: from DT-MRI to rule-based methods". Royal Society Open Science 7.4 (2020).

[10] K. Gillette et al. “A personalized real-time virtual model of whole heart electrophysiology”. Frontiers in Physiology (2022).

[11] L. Marx et al. “Personalization of electro-mechanical models of the pressure-overloaded left ventricle: fitting of windkessel-type afterload models” Philosophical Transactions of the Royal Society A: Mathematical, Physical and Engineering Sciences 378 (2020).

[12] M. Peirlinck et al. “Precision medicine in human heart modeling” Biomech. Model. Mechanobiol. 20 (2021)

Round 2

Reviewer 1 Report

Comments and Suggestions for Authors

Having read the revision of manuscript with the ID: mca-2251945, it is necessary to mention that some of my comments (minor ones) have been addressed in the revised version, but important ones still need to be done. I am drawing the attention of the authors to the following major comments of the current version of the manuscript.

1)      Levenberg-Marquardt algorithm is old optimization method based on an iterative procedure and initial guess would play in important role on results. The quality of the initial guess on this optimization method is even more influential on the convergence speed and the final solution compared to other optimizations methods. The manuscript lacks an explanation or discussion on these issues, and even the method also has not been described sufficiently.

2)      The authors cited their own manuscript that has not been concluded or finalized yet. The presentation or submission of an optimization method in two manuscripts, which is the most important feature of this research, remains on authors’ decisions and responsibilities for that matter.

3)      The alignments between simulation and Klotz methods are not good enough and I do not know why in section 3.4 it has been mentioned that Figs 4 and 5 show great agreements. I would suggest changing the R^2 criteria, but, if you want to use it you should continue on your trial procedure to achieve the level of 0.99 and more for the patients.  

4)      The unloaded shape of the ventricles has not been obtained yet, so it should be clearly mentioned in manuscript with explanation or estimation of the effects of this approximation.

5)      Figure No. 4 is not correct and the directions of the fibers in human heart in epicardium should not be obtained like that.

Reviewer 2 Report

Comments and Suggestions for Authors

The authors have made significant improvements to the manuscript, making it much clearer. They have addressed most of my concerns appropriately, except:

 "Acomp = 100 kPa has been chosen according to Rama et al. 2020. Acomp is a penalty parameter without actual physical meaning. Its value needs to be chosen high enough to ensure near incompressible material behaviour without causing numerical instability and unphysical behaviour (see e.g. Goektepe 2011 and Yin et al. 1996)."

This is clear, however, passive material parameters are highly sensitive to the chosen finite element formulation, including the penalty parameter A_comp, as, e.g., discussed in Section 3.3.4 of Marx et al. 2022 [1]

Furthermore, given that the authors use very coarse meshes (on average 3500 tetrahedral elements) combined with a simple linear finite element approach, it is likely that material parameters also show a high sensitivity to mesh resolution.

As the authors use the fitted passive parameters for their statistical analysis, I suggest that the authors test their method for sensitivity to (i) the finite element formulation (or at least the penalty parameter A_comp) and (ii) the mesh resolution as it was done e.g. in Finsberg et al. [2], Section 4.2.

In Finsberg et al. [2], the finite element approach with locking-free Taylor Hood elements was not very sensitive to mesh resolution, but it would be interesting and strengthen the paper if we could see a similar statement for the methods chosen in the present paper of Familusi et al.

[1] Marx et al., “Robust and efficient fixed-point algorithm for the inverse elastostatic problem to identify myocardial passive material parameters and the unloaded reference configuration" Journal of Computational Physics, vol. 463, 2022, DOI:10.1016/j.jcp.2022.111266

[2] Finsberg et al., “Efficient estimation of personalized biventricular mechanical function employing gradient-based optimization,” International Journal for Numerical Methods in Biomedical Engineering, vol. 34, p. e2982, 2018, DOI 10.1002/cnm.2982

Reviewer 3 Report

Comments and Suggestions for Authors

The authors have adequately addressed all the raised points, making it suitable for me to recommend its publication in the journal.

Author Response

We would like to thank the reviewer for the meticulous reading of the manuscript, the encouraging feedback, and the comments and suggestions which helped to improve the paper.

Best regards,

Sebastian Skatulla

Round 3

Reviewer 1 Report

Comments and Suggestions for Authors

Having read the response of the authors and the revision of the manuscript with the ID: mca-2251945, I realized that almost none of my important concerns have been addressed. 

There are some statements in the authors' response that are not correct. For example, if we consider this statement of the author:” The anisotropy coefficients ai i = 1 − 6 are usually not patient-specific due to the lack of combined strain and fibre orientation data”, then we have to change the objective of the study and the title of the manuscript as well. There are some other statements like “In the absence of patient-specific DT-MRI and full-field strain data linked to it, it is not possible to obtain a better fit which has been already stated under model limitations in Section 4.1.” which is not correct. 

My overall recommendation was to reconsider the manuscript after major revision and unfortunately, such revision has not been provided by authors. The limitations of this study are more than the scope of patient-specific modeling. Even if we change or reword the objectives and statements in the current version of the manuscript, my decision is to reject it.

Author Response

Please refer to attached PDF file.

Reviewer 2 Report

Comments and Suggestions for Authors

The authors addressed my concerns adequately

Author Response

Dear Reviewer

Many thanks again for your engagement with our manuscript helping us to improve it.

Regards,

Sebastian Skatulla